# Fast Incomplete Multi-view Clustering by Flexible Anchor Learning

**Yalan Qin** [1]  **Guorui Feng** [1]  **Xinpeng Zhang** [1]

## Abstract

Multi-view clustering aims to improve the final performance by taking advantages of complementary and consistent information of all views. In real world, data samples with partially available information are common and the issue regarding the clustering for incomplete multi-view data is inevitably raised. To deal with the partial data with large scales, some fast clustering approaches for incomplete multi-view data have been presented. Despite the significant success, few of these methods pay attention to learning anchors with high quality in a unified framework for incomplete multi-view clustering, while ensuring the scalability for large-scale incomplete datasets. In addition, most existing approaches based on incomplete multi-view clustering ignore to build the relation between anchor graph and similarity matrix in symmetric nonnegative matrix factorization and then directly conduct graph partition based on the anchor graph to reduce the space and time consumption. In this paper, we propose a novel fast incomplete multi-view clustering method for the data with large scales, termed Fast Incomplete Multi-view clustering by flexible anchor Learning (FIML), where graph construction, anchor learning and graph partition are simultaneously integrated into a unified framework for fast incomplete multi-view clustering. To be specific, we learn a shared anchor graph to guarantee the consistency among multiple views. The relation between anchor graph and similarity matrix in symmetric nonnegative matrix factorization can also be built. Experiments conducted on different datasets confirm the superiority of FIML compared with other clustering methods for incomplete multi-view data.

[1] School of Communication and Information Engineering, Shanghai University, Shanghai 200444, China. Correspondence to: Guorui Feng <fgr2082@aliyun.com>.

*Proceedings of the 42nd International Conference on Machine Learning*, Vancouver, Canada. PMLR 267, 2025. Copyright 2025 by the author(s).

## 1. Introduction

In real application, data are usually represented with different features from multiple views (Liu et al., 2025). This kind of data is usually named multi-view data and integrating the various information for clustering has shown to be a critical task in the unsupervised learning field. By investigating the complementary and diverse information among different views, a large number of clustering methods for multi-view data have been given (Zhang et al., 2021a; Kumar et al., 2011; Qin et al., 2022a; Chen et al., 2022; Zhao et al., 2023; Wang et al., 2023; Yu et al., 2023; Jia et al., 2023; Liu et al., 2024; Qin et al., 2023b; Liu et al., 2023b; Sun et al., 2024; Qin et al., 2024a; Liu et al., 2022a; Qin et al., 2024d; Liu et al., 2023a; Qin et al., 2025c; Liu et al., 2022b; Qin et al., 2025e) in recent years, which is different from the clustering for single view (Qin et al., 2023d;a; 2021; 2022b; 2025b; Pu et al., 2024). For instance, Kumar et al. (Kumar et al., 2011) guaranteed that different representations are able to agree with each other by co-regularizing the clustering hypotheses. Ye et al. (Ye et al., 2016) maximized the sum of weighted similarities among multiple clusterings to study the underlying clustering. Nie et al. (Nie et al., 2017) simultaneously learned the local structure as well as performed semi-supervised classification or clustering. Luo et al. (Luo et al., 2018) studied specificity and consistency in the representations from different views. Chen et al. (Chen et al., 2020) jointly explored the affinity matrix as well as the low-rank representation tensor. Zhou et al. (Zhou et al., 2020) utilized the predefined kernels to learn a consistent representation or a shared kernel and then achieved the unified clustering results. The vital part of clustering for multi-view data is to study the consistency of different views by learning a unified representation. Most existing multi-view clustering works make the assumption that data samples from different views are available (Zhao et al., 2017; Zhang et al., 2021b).

However, data samples in most applications often lack the information for some views (Xu et al., 2023; 2019; Qin et al., 2023c; Lv et al., 2022). Then the approaches based on integrity have difficulty in dealing with incomplete data from multiple views. In order to handle such problem, several methods of incomplete multi-view clustering have been presented (Zhang et al., 2020). We can conclude these

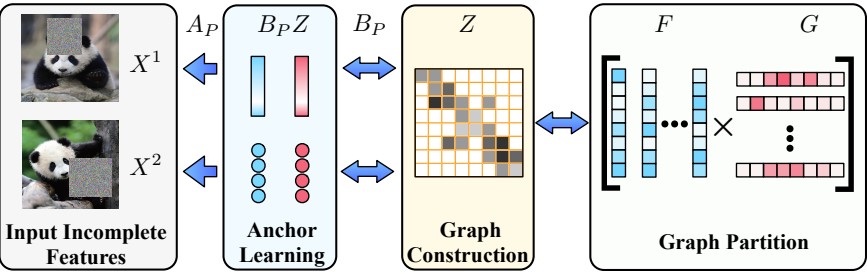

*Figure 1.* Framework of FIML. It jointly models graph construction, anchor learning and graph partition in a unified framework for fast incomplete multi-view clustering. To be specific, $X^1$ and $X^2$ are incomplete multi-view datasets as input, $B_p$ denotes the projection as the anchor guidance, $A_p$ is the indicator representation for the missing data, $Z$ refers to the shared anchor graph, $G$ and $F$ denote the centroid matrix and cluster assignment, respectively.

methods into three strategies including graph construction, matrix factorization and deep learning. The methods based on graph construction aim to produce a similarity matrix shared by different views. For instance, Liu et al. (Liu et al., 2017) simultaneously learned a representation and filled in the blank value. The methods based on matrix factorization make full use of $L_1$ constraint and nonnegative matrix factorization to learn a consensus representation (Li et al., 2014). Shao et al. (Shao et al., 2015) integrated weighted matrix factorization and $L_{2,1}$ regularization to obtain better clustering performance. The methods based on deep learning use a deep neural network to recover the missing data and the feature representation. As a representative, Lin et al. (Lin et al., 2021) employed contrastive learning for integrating data recovery and feature learning. However, most existing methods easily suffer from the high computation and space cost, which inevitably restricts their availability on the datasets with large scales.

To cope with the above issue, many methods for the data with large scales have been proposed (Qin et al., 2025a;d; Qin & Qian, 2024; Qin et al., 2024c;b). Wang et al. (Wang et al., 2011) built a constrained factor matrix for exploring the cluster structure. Kang et al. (Kang et al., 2020) employed $K$-means to obtain the anchors and then collocated them for a unified representation. Li et al. (Li et al., 2022) adopted the consistent learned anchors for handling the clustering problems of incomplete multi-view data. Nie et al. (Nie et al., 2020) simultaneously performed clustering on column and row of the original dataset. Wang et al. (Wang et al., 2022b) used the guidance of consensus anchors to study the anchor graph shared by different views. Sun et al. (Sun et al., 2021) exploited the underlying distribution of the data to construct the anchor graph. Among these existing methods, the approaches based on anchor have achieved attention due to the scalability and efficiency. This kind of methods usually employs the original data and the generated anchors to build the corresponding anchor graph, resulting in more satisfied clustering performance. Despite great success, the above methods neglect learning

high-quality anchors in a unified framework for incomplete multi-view clustering, while ensuring the scalability for incomplete datasets with large scales. In addition, few of the existing approaches based on incomplete multi-view clustering pay attention to building the relation between anchor graph and similarity matrix in symmetric nonnegative matrix factorization and then directly performing graph partition based on the anchor graph for reducing the space and computation consumption.

In this paper, we propose a novel fast incomplete multi-view clustering method for the data with large scales, termed Fast Incomplete Multi-view clustering by flexible anchor Learning (FIML), where graph construction, anchor learning and graph partition are simultaneously considered in a unified framework for fast incomplete multi-view clustering as Fig. 1. These three parts can boost each other, which promotes the quality of the clustering and improves the efficiency for large scale datasets. To be specific, we learn a shared anchor graph to guarantee the consistency among multiple views and adopt a adaptive weight coefficient to balance the impact for each view. The relation between anchor graph and similarity matrix in symmetric nonnegative matrix factorization can also be built, i.e., each entry in the anchor graph can describe the similarity between the anchor and original data sample. In particular, we constrain the factor matrix to be a cluster indicator representation by introducing the orthogonal constraint on the actual bases. Furthermore, we adopt the alternative algorithm for solving the optimization problem.

As a summary, the proposed FIML has the main contributions in the following:

- We give a new insight to the community of incomplete multi-view clustering for large scale datasets, i.e., graph construction, anchor learning and graph partition in fast incomplete multi-view clustering can boost each other, which are able to be integrated into a problem. The combination of these three issues is

the focus in our work. While most existing work treat graph construction, anchor learning and graph partition as separated problems in incomplete multi-view clustering for the datasets with large scales.

- We propose a novel fast incomplete multi-view clustering method for large scale data termed as FIML, where graph construction, anchor learning and graph partition are simultaneously considered in a unified framework for fast incomplete multi-view clustering. The relation between anchor graph and similarity matrix in symmetric nonnegative matrix factorization is also built, i.e., each entry in the anchor graph is able to characterize the similarity between the anchor and original data sample.

- Based on the relation between anchor graph and similarity matrix, we constrain the factor matrix with rigorous interpretation to be cluster indicator representation by introducing the orthogonal constraint on the actual bases and use the alternative algorithm for solving the formulated problem. Extensive experiments are performed on different datasets to demonstrate the superiority of FIML in terms of effectiveness and efficiency.

## 2. Fast Incomplete Multi-view Clustering by Flexible Anchor Learning

This section begins with introducing the motivation and formulation of the proposed FIML, then moves on to the detailed optimization process for FIML. We lastly conduct the analysis about the computation complexity to demonstrate the efficiency of FIML.

**Motivation:** For large-scale incomplete data clustering, reducing the redundancy of the data is the key to increase efficiency. Some existing works denote each data sample with a linear combination of the others and the global relation can be well exploited in this manner. However, the relatively high storage and computation time produced in this way inevitably limit the scalability of incomplete multi-view clustering for large-scale dataset. Actually, relatively less data samples are enough to reconstruct the latent space. Therefore, selecting some data samples from the original dataset as anchors or landmarks for reconstructing the relation structure is commonly used in the existing works.

Nevertheless, some existing incomplete multi-view clustering approaches usually conduct strategies based on heuristic sampling, where the anchor selection and graph construction are separated. Then the graph is constructed after selecting the anchors for different views. In this manner, the complementary information among different views is not able to be well explored and further algorithm is needed to obtain a shared graph. Afterwards, the clustering al-

gorithm (spectral clustering) is usually needed to achieve the final clustering results. This multiple-stage process significantly affects the quality of the anchors. Besides, few of the existing methods pay attention to building the relation between anchor and similarity matrix in symmetric nonnegative matrix factorization. As is known, each entry of a similarity matrix can describe the similarity between data samples in the dataset. Performing symmetric nonnegative matrix factorization for the similarity matrix can directly lead to the final partition. Then building the relation between anchor and similarity matrix can take advantages of directly obtaining the final clustering results in incomplete multi-view clustering. How to learn anchors with high quality in a unified framework and build the relation between anchor graph and similarity matrix in symmetric nonnegative matrix factorization to ensure the scalability on large-scale dataset for incomplete multi-view clustering remains a considerably challenging issue.

**Formulation:** Different from most existing works for incomplete multi-view clustering, we learn anchors instead of selecting them based on the available data samples in the dataset. The proposed FIML integrates graph construction, anchor learning and graph partition into a unified framework for fast incomplete multi-view clustering. Then the discriminative anchors are automatically learned and the final partition can be achieved in this manner. Based on the assumption that multiple views are sampled from a latent space, the anchors from multiple views are expected to be consistent in this space. Given multi-view dataset $\{X_p\}_{p=1}^v$, we construct the projection $B_p \in R^{d_p \times m}$ as the consensus anchor guidance to integrate complementary information from different views into the shared anchor graph $Z \in R^{m \times n}$, where $d_p$ and $m$ are the dimension of the data and the total number of anchors for the $p$-th view, respectively. The indicator representation $A_p \in \{0,1\}^{n \times n_p}$ is adopted to mark the unavailable data samples. The above process can be formulated as follows:

$$\min_{\alpha, Z, \{B_p\}_{p=1}^v} \sum_{p=1}^v \alpha_p^2 \|X_p A_p - B_p Z A_p\|_F^2,$$
$$s.t.\ \alpha^T \mathbf{1} = 1,\ B_p^T B_p = I,\ Z \geq 0,\ Z^T \mathbf{1} = 1, \tag{1}$$

where $\alpha_p^2$ is the coefficient of each view. It can be learned based on the contribution to the shared anchor graph. $X_p A_p$ denotes the available data samples for the $p$-th view. Since the space complexity of anchor graph $Z$ is $O(m \times n)$, we can relate $Z$ with similarity matrix in symmetric nonnegative matrix factorization for directly obtaining the final partition. As is known, symmetric nonnegative similarity matrix with the space complexity $O(n \times n)$ can be adopted to achieve the final clustering results based on factorization. Each entry in the anchor graph $Z$ describes the similarity between data sample and anchor. Since the symmetric constraint on $Z \in R^{m \times n}$ is not guaranteed in factorization

with $m \ll n$, we remove such constraint on anchor graph $Z$ and this is the main difference between anchor graph and similarity matrix in symmetric nonnegative matrix factorization. We then introduce the centroid matrix $G \in R^{m \times k}$ and the cluster assignment $F \in R^{k \times n}$ with $k$ being the total number of clusters in the dataset as follows:

$$\min_{G,F} \|Z - GF\|_F^2, \quad s.t. \ G^T G = I,$$

$$F_{ij} \in \{0,1\}, \quad \forall j = 1, 2, \cdots, n, \ \sum_{i=1}^{k} F_{ij} = 1, \tag{2}$$

where $F_{ij} = 0$ if the $j$-th data sample is not belonged to the $i$-th cluster and 1 otherwise. Note that imposing the orthogonal constraint on the actual bases can guide learning the factor matrix with rigorous clustering interpretation. To combine the partition information into the unified model, we formulate the total objective function as:

$$\min_{G,F,\alpha,Z,\{B_p\}_{p=1}^v} \sum_{p=1}^{v} \alpha_p^2 \|X_p A_p - B_p Z A_p\|_F^2 + \lambda \|Z - GF\|_F^2,$$

$$s.t. \ G^T G = I, \ \sum_{i=1}^{k} F_{ij} = 1, \ F_{ij} \in \{0,1\}, \ \forall j = 1, 2, \cdots, n,$$

$$\alpha^T \mathbf{1} = 1, \ B_p^T B_p = I, \ Z \geq 0, \ Z^T \mathbf{1} = 1, \tag{3}$$

where $\lambda > 0$ denotes the parameter for balancing different terms. Then graph construction, anchor learning and graph partition are jointly integrated into a unified framework for incomplete multi-view clustering in this manner, where these three parts can boost each other to achieve effective and efficient clustering results for incomplete large-scale multi-view dataset.

**Optimization:** We then design an alternating algorithm for optimizing each variable in Eq. (3) by fixing the others.

**(1) Optimize $\{B_p\}_{p=1}^v$**: With other variables being fixed, the objective function for $\{B_p\}_{p=1}^v$ can be rewritten as

$$\min_{\{B_p\}_{p=1}^v} \sum_{p=1}^{v} \alpha_p^2 \|X_p A_p - B_p Z A_p\|_F^2, \quad s.t. \ B_p^T B_p = I. \tag{4}$$

We can remove the irrelevant items and transform Eq. (4) into the form as follows:

$$\max_{B_p} Tr(B_p \Lambda_p), \quad s.t. \ B_p^T B_p = I, \tag{5}$$

where $\Lambda_p = (X_p \otimes H_p)Z^T$, $\otimes$ denotes the Hadamard product, $H_p = 1_{d_p} a_p$, $a_p = [a_{p,1}, \cdots, a_{p,n}]^T$ and $a_{p,j} = \sum_{l=1}^{n_p} A_{p,l,j}$. After conducting the singular value decomposition on $\Lambda_p$, the optimal solution of $B_p$ can be derived as $\Xi_m \Psi_m^T$, where $\Xi_m$ and $\Psi_m$ represent the matrices with

the first $m$ singular vectors of $\Lambda_p$ in the left and right, respectively.

**(2) Optimize $Z$** : With others being fixed, the objective for $Z$ turns to solve the problem as:

$$\min_{Z} \sum_{p=1}^{v} \alpha_p^2 \|X_p A_p - B_p Z A_p\|_F^2 + \lambda \|Z - GF\|_F^2,$$

$$s.t. \ Z \geq 0, \ Z^T \mathbf{1} = 1. \tag{6}$$

We then remove the irrelevant items and rewrite Eq. (6) as follows:

$$\min_{Z} \sum_{p=1}^{v} \alpha_p^2 Tr(Z^T Z(Q_p + \frac{\lambda}{\alpha_p^2} I)$$

$$- 2X_p^T B_p Z Q_p - 2\frac{\lambda}{\alpha_p^2} Z^T GF), \quad s.t. \ Z \geq 0, \ Z^T \mathbf{1} = 1, \tag{7}$$

where $Q_p = A_p A_p^T$. Since $z_i$ can be denoted as a vector with $z_{j,i}$ being the $j$-th entry, we can solve Eq. (7) by column as follows:

$$\min_{z_i} \|z_i - y_i\|_F^2, \quad s.t. \ z_i \geq 0, \ z_i^T \mathbf{1} = 1, \tag{8}$$

where $y_i^T = \sum_{p=1}^{v} \alpha_p^2 H_{p,i,j} X_{p,:,i}^T B_p / \lambda + \sum_{p=1}^{v} \alpha_p^2 H_{p,i,j}$. We then rewrite the Lagrangian function of Eq. (8) as:

$$L(z_i, \sigma_i, \gamma_i) = \|z_i - y_i\|_F^2 - \gamma_i^T z_i - \sigma_i(z_i^T 1 - 1), \tag{9}$$

where $\sigma_i$ and $\gamma_i$ correspond to Lagrangian multipliers. With KKT conditions, we have the equation:

$$\begin{cases} z_i - y_i - \sigma_i 1 - \gamma_i = 0 \\ \gamma_i \otimes z_i = 0. \end{cases} \tag{10}$$

Combining the constraint $z_i^T \mathbf{1} = 1$, we can obtain the equation as follows:

$$z_i = \max(y_i + \sigma_i 1, 0), \quad \sigma_i = \frac{1 + y_i^T 1}{m}. \tag{11}$$

**(3) Optimize $G$**: With other variables being fixed, the objective function for $G$ is transformed into the problem as follows:

$$\min_{G} \lambda \|Z - GF\|_F^2, \quad s.t. \ G^T G = I. \tag{12}$$

The optimization for $G$ can be written as

$$\max_{G} Tr(G^T J), \quad s.t. \ G^T G = I, \tag{13}$$

where $J = ZF^T$. Then the optimal solution for $G$ is $U_J V_J^T$ with $J = U_J \Sigma_J V_J^T$ based on singular value decomposition (SVD).

**(4) Optimize $F$**: With other variables being fixed, the objective function for $F$ can be formulated as the minimization problem as:

$$\min_{F} \lambda \|Z - GF\|_F^2,$$
$$s.t. \ F_{ij} \in \{0,1\}, \ \forall j = 1, \ 2, \ \cdots, \ n, \sum_{i=1}^{k} F_{ij} = 1, \quad (14)$$

We then have the minimization problem as

$$\min_{F_{:,j}} \lambda \|Z_{:,j} - GF_{:,j}\|^2, \quad s.t. \ F_{:,j} \in \{0,1\}^k, \ \|F_{:,j}\|_1 = 1. \quad (15)$$

Then the optimal row can be achieved by

$$i^* = \arg \min_{i} \|S_{:,j} - G_{:,i}\|^2. \quad (16)$$

According to Eq. (16), we can find that the optimal cluster assignment is achieved by the cluster centroid and the object.

**(5) Optimize $\alpha_p^v$**: With other variables being fixed, the objective function for $\alpha_p^v$ is:

$$\min_{\alpha} \sum_{p=1}^{v} \alpha_p^2 \kappa_p, \quad s.t. \ \alpha^T \mathbf{1} = 1, \ \alpha \geq 0, \quad (17)$$

where $\kappa_p = \|X_p A_p - B_p Z A_p\|_F^2$. We can obtain the optimal $\alpha$ based on Cauchy-Schwarz inequality as:

$$\alpha = \frac{\delta}{\sum_{p=1}^{v} \delta_p}, \quad (18)$$

where $\delta = [\delta_1, \ ..., \ \delta_v]$ with $\delta_p = 1/\kappa_p$.

### 2.1. Complexity Analysis

The computation burden of FIML consists of the optimization cost of each variable. To be specific, the time complexity for optimizing $B_p$ is $O(m^2 d + mnd)$ at each iteration. Optimizing the weight $\alpha$ of each view costs $O(mnd)$. The time cost to learn the shared anchor graph $Z$ is $O(mnd)$. For optimizing $F$, the time cost is $O(mnk)$. The time cost to update $G$ is $O(mk^2)$. Then the total time complexity of the proposed FIML is $O((m^2 d + mnd + mnk + mk^2)t)$, where $t$ denotes the number of iterations for these parts. Due to $n \gg m$ and $n \gg k$, the computation cost of FIML is nearly linear to the size of the dataset $O(n)$.

## 3. Experiments

In this section, we perform experiments to validate the effectiveness and efficiency of FIML on several widely used multi-view datasets. Among these datasets, there are some large-scale datasets for better verifying the clustering performance and running time of FIML.

---

**Algorithm 1** Algorithm of FIML

**Input:** Incomplete multi-view dataset $\{X_p^v\}_{p=1}^{v}$, the total number of clusters $k$ and the missing indicator $\{H_p^v\}_{p=1}^{v}$.
**Output:** The final cluster assignment $F$.
1: **Initialize:** Initialize $Z$, $\{B_p^v\}_{p=1}^{v}$ and $\{\alpha_p\}_{p=1}^{v}$.
2: **repeat**
3:     Update $Z$ by solving the problem in Eq. (6);
4:     Update $\{B_p^v\}_{p=1}^{v}$ by solving the problem in Eq. (4);
5:     Update $G$ by solving the problem in Eq. (12);
6:     Update $F$ by solving the problem in Eq. (14);
7:     Update $\alpha$ by solving the problem in Eq. (17);
8: **until** convergence

---

### 3.1. Compared Methods

The experiments are conducted on several widely adopted datasets including news groups (NGs), WebKB, ORL, STL10, MNIST and Cifar100. We compare FIML with some representative incomplete multi-view clustering approaches as follows: **BSV** (Ng et al., 2001), **MIC** (Shao et al., 2015), **MKKM-IK** (Ma et al., 2021), **DAIMIC** (Hu & Chen, 2018), **APMC** (Guo & Ye, 2019), **PIC** (Wang et al., 2019), **EEIMVC** (Liu et al., 2021), $V^3H$ (Fang et al., 2020), **IMVC-CBG** (Wang et al., 2022a), and **FIMVC-VIA** (Liu et al., 2022c).

In the experiment, we use four metrics to evaluate the experimental results, which include accuracy (ACC), NMI, F1-score and Purity. We repeat each algorithm for total 20 times and then report the mean and standard deviation of the results. The parameters for the compared methods of incomplete multi-view clustering are set as their recommended ones. We run all experiments on AMD Ryzen 5 Six-Core Processor 3.60 GHz.

### 3.2. Parameter Selection

There are total two parameters appeared in FIML, including the trade-off parameter $\lambda$ and the number of anchors $m$. We then perform experiments on different datasets to study how these two parameters influence the final clustering performance. We set $\lambda$ and $m$ in the range of $[0.001, 0.1, 1, 10, 100, 1000]$ and $[k, 2k, 3k, 5k, 7k]$, respectively. Here, $k$ corresponds to the total number of clusters in dataset. According to Figs. 2-3, we find that better performance is achieved when $\lambda = 1$ under the same $m$ on different datasets. Besides, the clustering result of FIML is relatively stable over different parameter values on these datasets, which shows that FIML is generally robust to the trade-off parameter $\lambda$. It can also be observed that different number of anchors $m$ has relatively little influence on the clustering performance under the same $\lambda$ for these datasets.

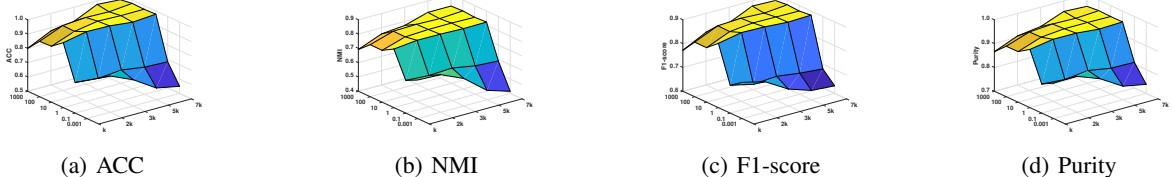

(a) ACC       (b) NMI       (c) F1-score       (d) Purity

*Figure 2.* Parameter Study of $m$ and $\lambda$ on NGs in terms of four metrics.

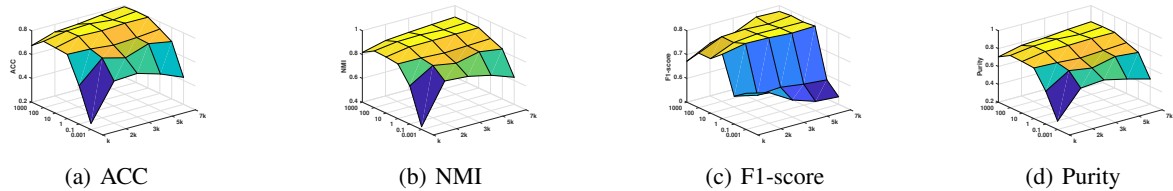

(a) ACC       (b) NMI       (c) F1-score       (d) Purity

*Figure 3.* Parameter Study of $m$ and $\lambda$ on ORL in terms of four metrics.

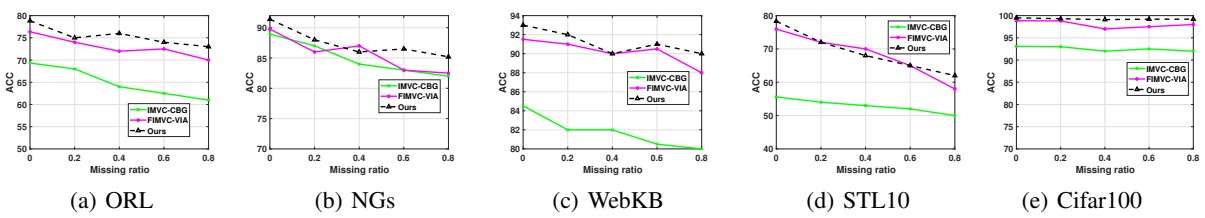

(a) ORL     (b) NGs     (c) WebKB     (d) STL10     (e) Cifar100

*Figure 4.* Clustering Performance in terms of ACC on datasets with different missing ratios.

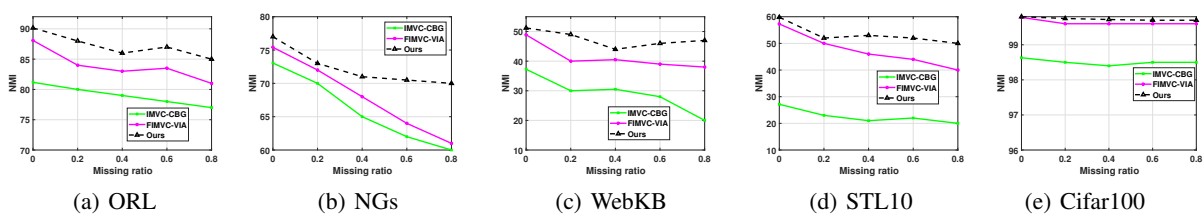

(a) ORL     (b) NGs     (c) WebKB     (d) STL10     (e) Cifar100

*Figure 5.* Clustering Performance in terms of NMI on datasets with different missing ratios.

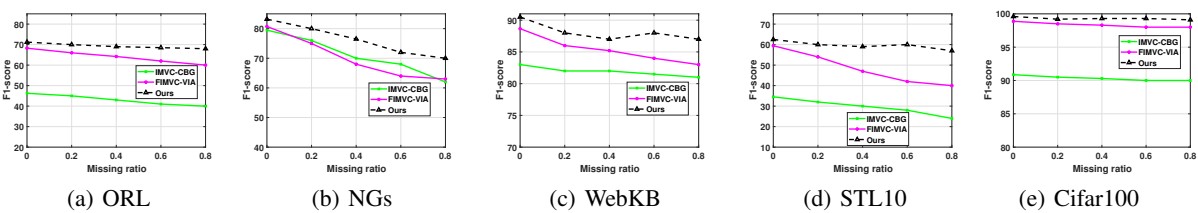

(a) ORL     (b) NGs     (c) WebKB     (d) STL10     (e) Cifar100

*Figure 6.* Clustering Performance in terms of F1-score on datasets with different missing ratios.

*Table 1.* Clustering Performance based on ACC (%) on datasets. "N/A " denotes out of memory.

| Dataset | BSV | MIC | MKKM-IK | DAIMC | APMC | PIC | EEIMVC | $V^3$ | IMVC-CBG | FIMVC-VIA | Ours |
|---|---|---|---|---|---|---|---|---|---|---|---|
| ORL | 24.30±0.50 | 37.60±1.50 | 59.90±2.00 | 68.00±2.30 | 65.50±1.60 | 69.00±1.50 | 73.20±2.40 | 67.00±1.30 | 69.50±2.00 | 76.30±2.70 | 78.84±0.50 |
| NGs | 41.20±2.00 | 21.20±0.50 | 80.20±0.00 | 89.50±0.05 | 89.40±0.05 | 82.00±0.20 | 77.90±0.15 | 79.90±0.40 | 88.90±0.15 | 89.70±0.05 | 91.40±0.40 |
| WebKB | 57.00±2.20 | 63.80±0.50 | 68.00±0.00 | N/A | 85.30±0.05 | 71.60±0.00 | 61.80±3.40 | 75.20±0.50 | 84.50±0.50 | 91.50±0.50 | 93.00±0.26 |
| STL10 | 11.20±0.10 | N/A | 75.80±0.30 | 23.00±1.50 | 27.00±0.50 | 28.80±0.20 | 46.70±2.30 | 18.50±0.50 | 55.60±0.80 | 76.00±0.30 | 78.30±0.60 |
| MNIST | N/A | N/A | N/A | 97.60±0.50 | N/A | N/A | N/A | N/A | 98.20±0.05 | 98.75±0.30 | 98.90±0.00 |
| Cifar100 | N/A | N/A | N/A | 89.68±0.50 | N/A | N/A | N/A | N/A | 93.00±1.20 | 98.90±0.60 | 99.50±0.26 |

*Table 2.* Clustering Performance based on NMI (%) on datasets. "N/A " denotes out of memory.

| Dataset | BSV | MIC | MKKM-IK | DAIMC | APMC | PIC | EEIMVC | $V^3$ | IMVC-CBG | FIMVC-VIA | Ours |
|---|---|---|---|---|---|---|---|---|---|---|---|
| ORL | 48.52±0.80 | 56.50±0.80 | 76.20±1.00 | 83.00±1.10 | 80.30±0.80 | 83.20±0.50 | 85.40±1.30 | 81.00±0.50 | 81.20±1.50 | 88.00±1.30 | 90.15±0.60 |
| NGs | 20.20±1.30 | 2.30±0.50 | 63.10±0.10 | 73.40±0.05 | 73.41±0.20 | 65.60±0.10 | 57.20±0.20 | 59.00±0.40 | 73.00±0.05 | 75.50±0.05 | 77.00±0.18 |
| WebKB | 1.85±0.80 | 3.30±0.50 | 4.00±0.10 | N/A | 47.90±0.20 | 1.70±0.00 | 3.50±0.50 | 23.60±1.00 | 37.20±0.15 | 48.90±0.20 | 51.20±0.50 |
| STL10 | 0.16±0.20 | N/A | 60.30±0.40 | 5.00±1.20 | 11.00±0.90 | 14.20±0.15 | 29.80±3.00 | 5.90±0.50 | 27.20±0.20 | 57.35±0.20 | 59.80±0.50 |
| MNIST | N/A | N/A | N/A | 93.90±0.50 | N/A | N/A | N/A | N/A | 94.90±0.10 | 96.20±0.30 | 97.30±0.10 |
| Cifar100 | N/A | N/A | N/A | 98.20±0.20 | N/A | N/A | N/A | N/A | 98.60±0.30 | 99.70±0.10 | 99.80±0.20 |

## 3.3. Experimental Results

We list the detailed clustering results of FIML and the compared approaches on different datasets in terms of four metrics in Tables 1-4. Note that N/A is adopted to indicate that the method suffers from the error due to out of memory. We also compare FIML with IMVC-CBG and FIMVC-VIA under different missing ratios on several datasets under different metrics. According to Tables 1-4 and Figs. 4-7, we draw the following conclusions:

- The proposed FIML can provide better performance than other compared methods for incomplete multi-view clustering in terms of different metrics. For instance, FIML gains a better clustering performance of 9.84% than PIC in terms of ACC on ORL, which shows that combining graph construction, anchor learning and graph partition in a unified framework of incomplete multi-view clustering is able to boost each other and result in effective clustering results.

- Compared with other methods for incomplete multi-view clustering, FIML shows better clustering performance with different missing ratios on several datasets under four metrics, which shows that the learned anchors for representing all data samples are relatively informative for these datasets and methods based on kernel or graph do not show the same satisfied performance.

- FIML produces more satisfied clustering performance than FIMVC-VIA on different datasets, showing that using the unified framework integrated by graph construction, anchor learning and graph partition can help achieving better cluster assignment matrix and this matrix can directly result in the final results.

## 3.4. Running Time

In this part, we show the running time of FIML and the compared approaches on different benchmark datasets. Based on Table 5, we have the observations as follows:

- Our FIML needs less running time than other methods for incomplete multi-view clustering on different datasets in terms of ACC, which indicates its efficiency for computation cost. Some other methods for comparison suffer from the memory issue on MNIST and Cifar100 based on the running time, which further shows the efficiency of our FIML.

- Some compared methods are able to consume less running time on some small dataset, i.e., EEIMVC uses less computation cost than our FIML on some datasets. However, these methods do not perform as well as ours when running on large-scale datasets. It can be explained by the fact that using a unified framework integrated by graph construction, anchor learning and graph partition can improve the efficiency for incomplete multi-view clustering, especially on the datasets with large scales .

- The dataset with larger dimensions tends to need more running time when these datasets are close in the size, i.e, IMVC-CBG needs more running time on Cifar100 than MNIST and the latter dataset has smaller dimension than the former dataset. As the size of dataset increases, our FIML and the compared methods usually consume more running time on different datasets.

## 3.5. Ablation Study

In this section, we perform ablation study to validate the superiority of adopting a unified framework integrated by graph construction, anchor learning and graph partition. In

*Table 3.* Clustering Performance based on F1-score (%) on datasets. "N/A " denotes out of memory.

| Dataset | BSV | MIC | MKKM-IK | DAIMC | APMC | PIC | EEIMVC | $V^3$ | IMVC-CBG | FIMVC-VIA | Ours |
|---|---|---|---|---|---|---|---|---|---|---|---|
| ORL | 9.00±0.50 | 17.50±1.00 | 46.30±2.30 | 56.80±2.60 | 50.50±2.40 | 57.70±1.30 | 63.50±2.90 | 55.00±1.50 | 46.30±3.00 | 68.20±3.00 | 71.20±0.50 |
| NGs | 32.30±1.00 | 32.80±0.20 | 68.70±0.00 | 80.30±0.05 | 80.40±0.60 | 72.80±0.20 | 64.00±0.20 | 65.60±0.40 | 79.50±0.05 | 80.80±0.00 | 83.20±0.70 |
| WebKB | 60.50±1.50 | 62.00±0.50 | 64.60±0.30 | N/A | 85.00±0.05 | 73.60±0.00 | 62.80±0.20 | 71.90±0.40 | 83.00±0.07 | 88.70±0.05 | 90.50±0.05 |
| STL10 | 11.70±0.05 | N/A | 57.80±0.30 | 13.20±1.80 | 18.60±1.20 | 21.40±0.10 | 29.90±2.00 | 17.05±0.50 | 34.60±0.07 | 59.90±0.00 | 62.50±0.50 |
| MNIST | N/A | N/A | N/A | 95.50±0.30 | N/A | N/A | N/A | N/A | 96.20±0.10 | 97.50±0.50 | 99.20±0.10 |
| Cifar100 | N/A | N/A | N/A | 90.50±0.50 | N/A | N/A | N/A | N/A | 91.90±0.50 | 99.00±0.50 | 99.60±0.20 |

*Table 4.* Clustering Performance based on Purity (%) on datasets. "N/A " denotes out of memory.

| Dataset | BSV | MIC | MKKM-IK | DAIMC | APMC | PIC | EEIMVC | $V^3$ | IMVC-CBG | FIMVC-VIA | Ours |
|---|---|---|---|---|---|---|---|---|---|---|---|
| ORL | 26.90±0.90 | 40.50±1.50 | 63.00±2.00 | 71.90±1.60 | 69.30±1.20 | 72.30±1.00 | 76.00±2.10 | 70.20±1.00 | 69.30±1.80 | 79.10±2.00 | 82.50±0.29 |
| NGs | 43.10±1.50 | 21.50±0.50 | 79.60±0.05 | 89.50±0.05 | 89.42±0.05 | 82.40±0.20 | 77.80±0.10 | 79.80±0.40 | 88.70±0.05 | 90.00±0.06 | 93.12±0.05 |
| WebKB | 78.20±0.20 | 78.24±0.60 | 78.40±0.05 | N/A | 90.15±0.08 | 78.20±0.40 | 78.18±0.30 | 91.70±3.00 | 84.60±0.05 | 91.60±0.20 | 94.10±0.20 |
| STL10 | 11.30±0.05 | N/A | 75.80±0.30 | 23.20±1.80 | 27.60±1.20 | 29.30±0.15 | 46.90±2.00 | 18.60±0.50 | 55.60±0.08 | 76.00±0.20 | 78.90±0.55 |
| MNIST | N/A | N/A | N/A | 97.50±0.30 | N/A | N/A | N/A | N/A | 98.00±0.10 | 98.50±0.50 | 99.20±0.10 |
| Cifar100 | N/A | N/A | N/A | 92.50±0.50 | N/A | N/A | N/A | N/A | 94.90±0.50 | 99.00±0.50 | 99.55±0.20 |

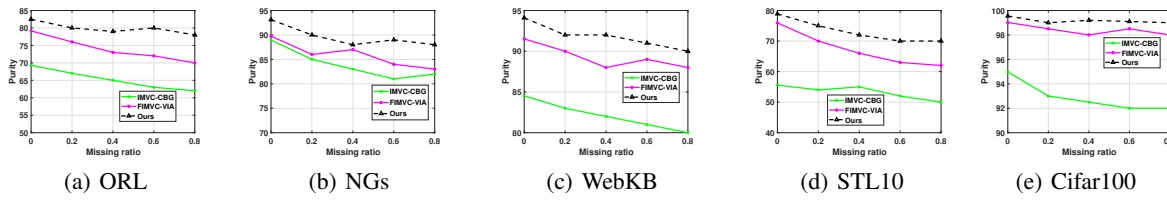

| (a) ORL | (b) NGs | (c) WebKB | (d) STL10 | (e) Cifar100 |
|---|---|---|---|---|

*Figure 7.* Clustering Performance in terms of Purity on datasets with different missing ratios.

*Table 5.* Running time of all methods on different datasets. "N/A " denotes out of memory.

| Dataset | BSV | MIC | MKKM-IK | DAIMC | APMC | PIC | EEIMVC | $V^3$ | IMVC | FIMVC | Ours |
|---|---|---|---|---|---|---|---|---|---|---|---|
| ORL | 0.15 | 425.00 | 0.50 | 1200.00 | 0.50 | 0.30 | 0.55 | 90.00 | 3.00 | 1.70 | 0.30 |
| NGs | 0.05 | 145.00 | 0.50 | 0.20 | 0.28 | 0.25 | 0.15 | 14.50 | 1.50 | 0.30 | 0.25 |
| WebKB | 0.15 | 340.50 | 3.20 | N/A | 1.20 | 0.22 | 32.00 | 0.65 | 0.28 | 0.24 | |
| STL10 | 66.90 | N/A | 1666.00 | 590.00 | 72.00 | 3350.00 | 68.50 | 45290.00 | 18.50 | 6.20 | 5.50 |
| MNIST | N/A | N/A | N/A | 5600.20 | N/A | N/A | N/A | N/A | 552.00 | 20.20 | 18.40 |
| Cifar100 | N/A | N/A | N/A | 25200.00 | N/A | N/A | N/A | N/A | 815.00 | 47.00 | 35.00 |

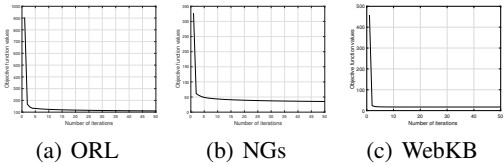

| (a) ORL | (b) NGs | (c) WebKB |
|---|---|---|

*Figure 8.* Convergence curve on different datasets. (a) ORL. (b) NGs. (c) WebKB.

*Table 6.* Ablation study based on separated or unified manner

| Metrics | Manner | ORL | NGs | WebKB | STL10 | MNIST | Cifar100 |
|---|---|---|---|---|---|---|---|
| ACC | Separated | 70.60±0.20 | 82.45±0.30 | 82.00±0.70 | 71.40±0.55 | 84.60±0.00 | 90.40±0.45 |
| | Unified | 78.84±0.50 | 91.40±0.40 | 93.00±0.26 | 78.30±0.60 | 98.90±0.00 | 99.50±0.26 |
| NMI | Separated | 75.20±0.15 | 70.39±0.05 | 44.60±0.78 | 48.20±0.27 | 92.00±0.70 | 91.30±0.09 |
| | Unified | 90.15±0.60 | 77.00±0.18 | 51.20±0.50 | 59.80±0.50 | 97.30±0.10 | 99.80±0.20 |
| F1-score | Separated | 62.49±1.00 | 76.20±0.30 | 80.20±0.60 | 54.90±0.15 | 90.50±0.90 | 90.49±0.55 |
| | Unified | 71.20±0.50 | 83.20±0.70 | 90.50±0.05 | 62.50±0.50 | 99.20±0.10 | 99.60±0.20 |
| Purity | Separated | 70.85±0.39 | 84.20±0.64 | 82.70±0.20 | 69.40±0.90 | 88.50±0.05 | 82.40±0.19 |
| | Unified | 82.50±0.29 | 93.12±0.05 | 94.10±0.20 | 78.90±0.55 | 99.20±0.10 | 99.55±0.20 |

### 3.6. Convergence Analysis

We conduct convergence analysis of FIML on different datasets by showing the evolution process of the objective function with iterations in terms of ACC. According to Fig. 8, we observe that FIML monotonically decreases with iterations and tends to converge in about some iterations on these datasets, which demonstrates the convergence of FIML.

## 4. Conclusion

we propose FIML in this work for fast incomplete multiview clustering. It simultaneously considers graph construction, anchor learning and graph partition in a unified framework, in which these parts boost each other for improving the effectiveness and efficiency for datasets with large scales. To be specific, a shared anchor graph for guar-

comparative experiments, we first learn anchors and construct the graph to obtain informative representation. Then the graph partition is isolated from the above two processes in the designed experiment. According to Table 6, we can find that the clustering performance of the proposed FIML significantly outperforms than the case in separated manner, demonstrating the necessary of using a unified framework integrated by graph construction, anchor learning and graph partition to directly achieve the final assignment.

anteeing the consistency among multiple views is learned and the adaptive weight coefficient is adopted to balance the impact for each view. We then adopt the alternative algorithm to solve the optimization problem. Extensive experiments on several benchmark datasets show the effectiveness and efficiency of FIML under different metrics.

## Acknowledgments

The work was partially supported by Eastern Talent Plan Leading Project under Grant BJKJ2024011 and National Natural Science Foundation of China (62402303).

## Impact Statement

This paper presents work whose goal is to advance the field of Machine Learning. There are many potential societal consequences of our work, none which we feel must be specifically highlighted here.

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
