# OpenReview forum: "Fast Incomplete Multi-view Clustering by Flexible Anchor Learning"
_ICML.cc/2025/Conference — ICML 2025 poster_

### Official Review · Reviewer_y35w · 2025-03-08

**Overall Recommendation:** 4

**Summary:**

This paper proposes FIML in this work for fast incomplete multi-view clustering. It simultaneously considers graph construction, anchor learning and graph partition in a unified framework, in which these parts boost each other for improving the effectiveness and efficiency for datasets with large scales. To be specific, a shared anchor graph for guaranteeing the consistency among multiple views is learned and the adaptive weight coefficient is adopted to balance the impact for each view.

**Claims And Evidence:**

Yes.

**Essential References Not Discussed:**

No.

**Experimental Designs Or Analyses:**

Yes. The adopted experimental methods for comparison contain methods from the recent years, which enhances the credibility of the experimental performance.

**Methods And Evaluation Criteria:**

Yes.

**Other Comments Or Suggestions:**

No.

**Other Strengths And Weaknesses:**

Strength: The authors learn a shared anchor graph for guaranteeing the consistency among multiple views and adopt the adaptive weight coefficient to balance the impact for each view.

Weakness:
1. After giving the total objective function for the proposed FIML, the authors should show the summarization of this objective function after Eq. (3) to make the formulation of FIML better explained.
2. The authors should further what is KKT condition before Eq. (10) for the Optimization part of the proposed FIML.

3.The compared methods in the experiment are not enough and the authors are expected to add more recent works for comparison in Experiment part.

4. The length of motivation part in the work is oversized in Section 2 for the paper. Then the authors should highlight the most important part in this part to make the motivation more obvious.

5. The formats of works in references part should be consistent to increase the presentation of this paper.

**Questions For Authors:**

1. The authors are expected to add more recent works for comparison in Experiment part.

2. The authors should highlight the most important part in motivation part to make the motivation more obvious.

**Relation To Broader Scientific Literature:**

Compared to the existing works, this paper proposes FIML in this work for fast incomplete multi-view clustering. It simultaneously considers graph construction, anchor learning and graph partition in a unified framework, in which these parts boost each other for improving the effectiveness and efficiency for datasets with large scales.

**Theoretical Claims:**

Yes. There are no proofs for theoretical claims in this work and I do not need to check them.

---

> ### Author Rebuttal · Authors · 2025-03-28
>
> Q1: The summarization of the objective function of the proposed FIML after Eq. (3) should be given.
>
> A1: Thanks for the comment! The related summarization of the objective function for the proposed FIML after Eq. (3) is shown in the following. Then graph construction, anchor learning and graph partition are jointly integrated into a unified framework for incomplete multi-view clustering in this manner, where these three parts can boost each other to achieve effective and efficient clustering results for incomplete large-scale multi-view dataset. The discriminative anchors can be automatically learned and the final partition is achieved in this manner. We will add the above summarization for the camera-ready version.
>
>
> Q2: What is KKT condition before Eq. (10)?
>
> A2: Good question! It is needed to explain KKT and the meaning is not clear without the related description. KKT is the abbreviation for Karush-Kuhn-Tucker conditions and we will add this explanation for the camera-ready version.
>
> Q3: More recent methods should be added for comparison in the experiment.
>
> A3: Thanks for the comment! We have added a recent method for comparison in the experimental section, i.e., OMVCDR [a].
>
> [a] One-Step Multi-View Clustering With Diverse Representation, 2024
>
> The clustering results of OMVCDR based on ACC for all datasets are 77.00±0.00, 89.85±0.10, 91.75±0.00, 76.50±0.15, 98.80±0.00, 98.95±0.10
>
> The clustering results of OMVCDR based on NMI for all datasets are 89.50±0.00, 75.85±0.15, 49.00±0.05, 58.00±0.00, 96.50±0.10, 99.72±0.00
>
> The clustering results of OMVCDR based on F1-score for all datasets are 69.00±0.00, 81.15±0.10, 88.95±0.00, 60.00±0.05, 97.80±0.00, 99.25±0.00
>
> The clustering results of OMVCDR based on Purity for all datasets are 79.56±0.00, 91.25±0.20, 92.00±0.00, 76.50±0.50, 98.79±0.00, 99.20±0.05
>
> Q4: The most important part in motivation should be highlighted.
>
> A4: Good question! It is needed to highlight the most important part in motivation, which is presented as “Actually, relatively less data samples are enough to reconstruct the latent space. Therefore, selecting some data samples from the original dataset as anchors or landmarks for reconstructing the relation structure is commonly used in the existing works.” We will highlight the above details for the camera-ready version.
>
> Q5: The formats of works in references should be consistent.
>
> A5: Thanks for the comment! We will check the formats of works in references part for consistency in increasing the presentation of the paper.

---

### Official Review · Reviewer_dTzz · 2025-03-08

**Overall Recommendation:** 4

**Summary:**

This paper proposes a novel fast incomplete multi-view clustering method for the data with large scales, termed Fast Incomplete Multi-view clustering via flexible anchor Learning (FIML), where graph construction, anchor learning and graph partition are simultaneously integrated into a unified framework for efficient incomplete multi-view clustering. To be specific, the authors learn a shared anchor graph to guarantee the consistency among multiple views and employ a adaptive weight coefficient to balance the impact for each view. The relation between anchor graph and similarity matrix in symmetric nonnegative matrix factorization can also be built, i.e., each entry in the anchor graph can characterize the similarity between the anchor and original data sample.

**Claims And Evidence:**

Yes

**Essential References Not Discussed:**

No

**Experimental Designs Or Analyses:**

Yes. The used compared methods include the works from the recent years, increasing the credibility of the final results.

**Methods And Evaluation Criteria:**

Yes

**Other Comments Or Suggestions:**

No

**Other Strengths And Weaknesses:**

**Strength**

The authors constrain the factor matrix with rigorous interpretation to be cluster indicator representation by introducing the orthogonal constraint on the actual bases and use the alternative algorithm for solving the formulated problem. Extensive experiments are performed on different datasets to demonstrate the superiority of FIML in terms of effectiveness and efficiency.

**Weakness**

1. The authors state that each entry in the anchor graph $ Z $ describes the similarity between data sample and anchor. Since the symmetric constraint on $ Z\in R^{m\times n} $ are not guaranteed in factorization with $ m\ll n $, the authors remove such constraint on anchor graph $ Z $ and this is the main difference between anchor graph and similarity matrix in symmetric nonnegative matrix factorization. In this part, the authors are expected to give the reason why each entry in the anchor graph $ Z $ describes the similarity between data sample and anchor.
2. The authors list the detailed clustering results of FIML and the compared approaches on different datasets in terms of four metrics in Tables 1-4. The authors also compare FIML with IMVC-CBG and FIMVC-VIA under different missing ratios on several datasets under different metrics. According to Tables 1-4 and Figs. 4-7, the authors then draw some following conclusions. However, the authors do not bold the best clustering performance in Tables 1-4 in terms of four metrics for this paper.
3. The authors perform the parameter selection for the trade-off parameter $ \lambda $ in the range of $ [0.001,0.1,1,10,100,1000] $ to study how these this parameter influences the final clustering performance and find that better performance is achieved when $ \lambda=1 $ under the same $ m $ on different datasets. Besides, the clustering result of FIML is relatively stable over different parameter values on these datasets, which shows that FIML is generally robust to the trade-off parameter $ \lambda $. However, the authors do not give the detailed reason why the $ [0.001,0.1,1,10,100,1000] $ is chosen for parameter selection in this paper.
4. The authors perform ablation study to validate the superiority of adopting a unified framework integrated by graph construction, anchor learning and graph partition. In comparative experiments, the authors first learn anchors and construct the graph to obtain informative representation. Then the graph partition is isolated from the above two processes in the designed experiment. However, the authors do not give the detailed specific values analysis regarding the superiority of adopting a unified framework integrated by graph construction, anchor learning and graph partition.

**Questions For Authors:**

1. The authors conduct convergence analysis of FIML on different datasets by showing the evolution process of the objective function with iterations in terms of ACC and observe that FIML monotonically decreases with iterations and tends to converge in about some iterations on these datasets. Here, how many iterations are needed for the proposed FIML in reaching convergence?
2. The authors show the running time of FIML and the compared approaches on different benchmark datasets in Table 5. However, the authors do not give the memory description of the used device. What is the memory value of the used device?

**Relation To Broader Scientific Literature:**

This paper gives a new insight to the community of incomplete multi-view clustering for large scale datasets compared to the existing methods, i.e., graph construction, anchor learning and graph partition in efficient incomplete multi-view clustering can boost each other, which are able to be integrated into a problem. The combination of these three issues is the focus in our work. While most existing work treat graph construction, anchor learning and graph partition as separated problems in incomplete multi-view clustering for the datasets with large scales.

**Theoretical Claims:**

Yes. There exist no proofs for theoretical claims in the paper and it is not needed to check them.

---

> ### Author Rebuttal · Authors · 2025-03-28
>
> Q1: The reason why each entry in the anchor graph describes the similarity between data sample and anchor.
>
> A1: Thanks for the comment! The reason why each entry in the anchor graph Z describes the similarity between data sample and anchor can be explained by the fact that dimension of the anchor graph Z is m \times n. It corresponds to m anchors and n data samples and larger entry in the anchor graph Z indicates larger similarity between data sample and anchor. We will add the above explanations for the camera-ready version.
>
> Q2: The authors do not bold the best results in the experiment.
>
> A2: Good question! It is needed to bold the best clustering results on tables for the experiment to make the performance achievements more obvious. We will bold the best clustering results on tables for the camera-ready version in the experiment to make the performance achievements more obvious.
>
> Q3: The reason why the [0.001, 0.1, 1, 10, 100, 1000] is chosen for parameter selection should be given.
>
> A3: Thanks for the comment! The reason why the [0.001, 0.1, 1, 10, 100, 1000] is chosen for parameter selection is that values in such range represent different representative magnitudes in parameter selection for \lambda in the experiment. We will add this explanation for the camera-ready version.
>
> Q4: Giving the detailed specific values analysis for the experimental results.
>
> A4: Good question! It is needed to give the detailed specific values analysis regarding the superiority of adopting a unified framework integrated by graph construction, anchor learning and graph partition. On the STL10 dataset, our method outperforms the last five multi-view clustering methods in tables by achieving improvements of 49.5%, 31.6%, 59.8%, 22.7% and 2.3%. We will add the above detailed specific values analysis for the experimental results in the camera-ready version.
>
> Q5: How many iterations are needed for the proposed FIML in reaching convergence?
>
> A5: Thanks for the comment! It needs about 20 iterations for the proposed FIML in reaching convergence. We will add this description regarding the iteration convergence for the camera-ready version.
>
> Q6: What is the memory value of the adopted device in experiment?
>
> A6: Good question! It is needed to give the memory description of the adopted device considering the running time of FIML and the compared approaches on different benchmark datasets are shown in Table 5. The memory value of the used device is 8G and we will add this description for the camera-ready version.

---

> > ### Comment · Reviewer_dTzz · 2025-04-03
> >
> > Thanks for your responses. They have addressed my concerns.

---

### Official Review · Reviewer_Tzg8 · 2025-03-08

**Overall Recommendation:** 3

**Summary:**

This paper proposes a novel fast incomplete multi-view clustering method for the data with large scales, termed Fast Incomplete Multi-view clustering by flexible anchor Learning (FIML), where graph construction, anchor learning and graph partition are simultaneously considered in a unified framework for fast incomplete multi-view clustering. These three parts can boost each other, which promotes the quality of the clustering and improves the efficiency for large scale datasets. To be specific, the authors learn a shared anchor graph to guarantee the consistency among multiple views and adopt a adaptive weight coefficient to balance the impact for each view.

**Claims And Evidence:**

Yes.

**Essential References Not Discussed:**

No.

**Experimental Designs Or Analyses:**

Yes. the experimental designs are reasonable.

**Methods And Evaluation Criteria:**

Yes. The effectiveness of the proposed FIML is better demonstrated in the experiment.

**Other Comments Or Suggestions:**

No.

**Other Strengths And Weaknesses:**

Strength:

This paper learns a shared anchor graph to guarantee the consistency among multiple views and adopt a adaptive weight coefficient to balance the impact for each view. The relation between anchor graph and similarity matrix in symmetric nonnegative matrix factorization can also be built.

Weakness:

1. The authors propose a novel fast incomplete multi-view clustering method for the data with large scales, termed Fast Incomplete Multi-view clustering by flexible anchor Learning (FIML). The authors are expected to compare the proposed FIML with the most related works in Introduction part. Then the novelty of the proposed FIML is more clear for the readers.
2. The authors should add more recent related works in multi-view clustering for comparison in the experiment. Then the effectiveness of the proposed FIML is better demonstrated in the experiment.
3. Considering the running time is given in the experiment, the authors should give the memory the used device in the experiment.
4. The authors should bold the best and highlight the second best clustering performance for tables in the experiment to make the performance gains more obvious.
5. The authors are expected to give more detailed running time description in the related parts, i.e., what is the unit of running time in the experiment , second (s) or log second?

**Questions For Authors:**

1. The authors should give more detailed running time description in the related parts, i.e., what is the unit of running time in the experiment , second (s) or log second?
2. The authors are expected to add more recent related works in multi-view clustering for comparison in the experiment. Then the effectiveness of the proposed FIML is better demonstrated in the experiment.

**Relation To Broader Scientific Literature:**

The authors propose a novel fast incomplete multi-view clustering method for the data with large scales, termed Fast Incomplete Multi-view clustering by flexible anchor Learning (FIML), where graph construction, anchor learning and graph partition are simultaneously considered in a unified framework for fast incomplete multi-view clustering.

**Theoretical Claims:**

Yes. The authors do not give proofs for theoretical claims in this work and there is no need to check them.

---

> ### Author Rebuttal · Authors · 2025-03-28
>
> Q1: Compare the proposed FIML with the most related works in Introduction.
>
> A1: Thanks for the comment! The existing works most related to FIML are FPMVS-CAG and SMVSC. FPMVS-CAG jointly performs anchor selection and subspace graph construction into a framework. Then the two processes can be negotiated with each other to improve the clustering performance. SMVSC integrates anchor learning and graph construction into a unified optimization process. A more discriminative clustering structure can be achieved in this manner. The connection between the above two works and ours is that anchor learning and subspace graph construction are simultaneously conducted in a unified framework. The differences between these two works and ours, are that we learn a shared anchor graph to guarantee the consistency among multiple views and employ a adaptive weight coefficient to balance the impact for each view. The relation between anchor graph and similarity matrix in symmetric nonnegative matrix factorization can also be built, i.e., each entry in the anchor graph can characterize the similarity between the anchor and original data sample.
>
> Q2: Add more recent compared methods in the experiment.
>
> A2: Good question! We have added a recent method for comparison in the experimental section, i.e., OMVCDR [a].
>
> [a] One-Step Multi-View Clustering With Diverse Representation, 2024
>
> The clustering results of OMVCDR based on ACC for all datasets are 77.00±0.00, 89.85±0.10, 91.75±0.00, 76.50±0.15, 98.80±0.00, 98.95±0.10
>
> The clustering results of OMVCDR based on NMI for all datasets are 89.50±0.00, 75.85±0.15, 49.00±0.05, 58.00±0.00, 96.50±0.10, 99.72±0.00
>
> The clustering results of OMVCDR based on F1-score for all datasets are 69.00±0.00, 81.15±0.10, 88.95±0.00, 60.00±0.05, 97.80±0.00, 99.25±0.00
>
> The clustering results of OMVCDR based on Purity for all datasets are 79.56±0.00, 91.25±0.20, 92.00±0.00, 76.50±0.50, 98.79±0.00, 99.20±0.05
>
> Q3: The memory of the adopted device should be given.
>
> A3: Thanks for the comment! Since the running time is listed in the experiment, we should give the memory of the used device in the experiment. The memory of the adopted device in the experiment is 8G in the experiment and we will add this description for the camera-ready version.
>
> Q4: The best and second best results should be bold in the experiment.
>
> A4: Good question! It is important to bold the best and highlight the second best clustering performance for tables in the experiment in making the performance gains more obvious for the readers. We will bold the best and highlight the second best clustering performance for tables of the camera-ready version in the experiment to make the performance gains more obvious.
>
> Q5: What is the unit of running time in the experiment, second or log second?
>
> A5: Thanks for the comment! It is needed to give more detailed running time description in the related parts. The unit of running time in the experiment is log second and we will add this description for the camera-ready version.

---

### Official Review · Reviewer_agLA · 2025-03-10

**Overall Recommendation:** 4

**Summary:**

This paper proposes a novel fast incomplete multi-view clustering method for large scale data termed as FIML, where graph construction, anchor learning and graph partition are simultaneously considered in a unified framework for fast incomplete multi-view clustering. The relation between anchor graph and similarity matrix in symmetric nonnegative matrix factorization is also built, i.e., each entry in the anchor graph is able to characterize the similarity between the anchor and original data sample.

**Claims And Evidence:**

Yes.

**Essential References Not Discussed:**

No.

**Experimental Designs Or Analyses:**

Yes. The dataset used in the experiment covers a wide range of categories and quantities, enabling a comprehensive presentation for the final results. Additionally, the comparison of experimental methods consists of advanced models from the past three years, enhancing the credibility of the experimental results.

**Methods And Evaluation Criteria:**

Yes.

**Other Comments Or Suggestions:**

No.

**Other Strengths And Weaknesses:**

Strength: This paper builds the relation between anchor graph and similarity matrix in symmetric nonnegative matrix factorization, i.e., each entry in the anchor graph is able to characterize the similarity between the anchor and original data sample.

Weakness:

1. The authors should add more recent fast multi-view clustering methods for comparison in the experiment to validate the novelty of this work.

2. The authors can give more detailed experimental analysis for the experimental results based on ACC, NMI, F1-score and Purity in Table 1-Table 4.

3. The best clustering performance based on ACC, NMI, F1-score and Purity in Table 1-Table 4 in the experiment should be bold.

4. The authors should give the brief optimization process before the following optimization steps in Optimization part. Then the whole optimization routine becomes more clear.

**Questions For Authors:**

1. The authors should give more detailed experimental analysis for the experimental results based on ACC, NMI, F1-score and Purity in Table 1-Table 4.
2. The authors are expected to give the brief optimization process before the following optimization steps in Optimization part. Then the whole optimization routine becomes more clear.

**Relation To Broader Scientific Literature:**

Compared to the previous studies, the highlights of this research constrain the factor matrix with rigorous interpretation to be cluster indicator representation by introducing the orthogonal constraint on the actual bases and use the alternative algorithm for solving the formulated problem. Extensive experiments are performed on different datasets to demonstrate the superiority of FIML in terms of effectiveness and efficiency.

**Theoretical Claims:**

Yes. There are no proofs for theoretical claims and there is no need to check the correctness.

---

> ### Author Rebuttal · Authors · 2025-03-28
>
> Q1: The authors should add more recent works for comparison in the experiment.
>
> A1: Thanks for the comment! We have added a recent method for comparison in the experimental section, i.e., OMVCDR [a].
>
> [a] One-Step Multi-View Clustering With Diverse Representation, 2024
>
> The clustering results of OMVCDR based on ACC for all datasets are 77.00±0.00, 89.85±0.10, 91.75±0.00, 76.50±0.15, 98.80±0.00, 98.95±0.10
>
> The clustering results of OMVCDR based on NMI for all datasets are 89.50±0.00, 75.85±0.15, 49.00±0.05, 58.00±0.00, 96.50±0.10, 99.72±0.00
>
> The clustering results of OMVCDR based on F1-score for all datasets are 69.00±0.00, 81.15±0.10, 88.95±0.00, 60.00±0.05, 97.80±0.00, 99.25±0.00
>
> The clustering results of OMVCDR based on Purity for all datasets are 79.56±0.00, 91.25±0.20, 92.00±0.00, 76.50±0.50, 98.79±0.00, 99.20±0.05
>
> Q2: The authors should give more detailed analysis for the experimental results based on ACC, NMI, F1-score and Purity.
>
> A2: Good question! We have added more related detailed analysis for the experimental results as shown in the following. On the STL10 dataset, our method outperforms the last five multi-view clustering methods in tables by achieving improvements of 49.5%, 31.6%, 59.8%, 22.7% and 2.3%. We can also find that the anchor based algorithms are capable of handling the bigger data compared with the traditional multi-view clustering methods.
>
> Q3: The best clustering results in the experiment should be bold.
>
> A3: Thanks for the comment! We will bold the best clustering performance based on ACC, NMI, F1-score and Purity in Table 1-4 in the experiment for the camera-ready version to make the performance gains more obvious.
>
> Q4: The brief optimization process before the detailed steps should be given.
>
> A4: Good question! The algorithm of FIML consists of the steps shown in the following. It contains the input, output and initialization of the algorithm. The detailed optimization steps regarding each variable are also given in this algorithm. Then the optimization stops when the convergence condition achieves. We will add such brief optimization process before the detailed steps in Optimization part to make the whole optimization routine more clear.

---

> > ### Comment · Reviewer_agLA · 2025-04-07
> >
> > The authors have been well addressed my concerns, and I keep my score.

---

### Decision · Program_Chairs · 2025-05-01

**Decision:**

Accept (poster)

**Comment:**

This paper presents a fast incomplete multi-view clustering method by flexible anchor learning. Graph construction, anchor learning and graph partition are simultaneously integrated into a unified framework for fast incomplete multi-view clustering. Experiments on several datasets demonstrate the superiority. After the rebuttal, it receives three accept and one weak accept. Its merits, including the interesting idea, extensive experiments, and good results, are well recognized by the reviewers. The response well addresses the reviewers' concerns about the additional comparison, detailed analysis, and so on. I think the current manuscript meets the requirement of this top conference and recommend for acceptance. Please incorporate the revision in the updated manuscript.